# Spinal Canal and Spinal Cord in Rat Continue to Grow Even after Sexual Maturation: Anatomical Study and Molecular Proposition

**DOI:** 10.3390/ijms232416076

**Published:** 2022-12-16

**Authors:** Akihito Sotome, Ken Kadoya, Yuki Suzuki, Norimasa Iwasaki

**Affiliations:** Department of Orthopaedic Surgery, Faculty of Medicine and Graduate School of Medicine, Hokkaido University, Kita-15 Nishi-7, Kita-ku, Sapporo 060-8638, Japan

**Keywords:** spinal canal, spinal cord, the space available for the cord, growth curve, cervical spondylotic myelopathy, animal model, CT myelography

## Abstract

Although rodents have been widely used for experimental models of spinal cord diseases, the details of the growth curves of their spinal canal and spinal cord, as well as the molecular mechanism of the growth of adult rat spinal cords remain unavailable. They are particularly important when conducting the experiments of cervical spondylotic myelopathy (CSM), since the disease condition depends on the size of the spinal canal and the spinal cord. Thus, the purposes of the present study were to obtain accurate growth curves for the spinal canal and spinal cord in rats; to define the appropriate age in weeks for their use as a CSM model; and to propose a molecular mechanism of the growth of the adult spinal cord in rats. CT myelography was performed on Lewis rats from 4 weeks to 40 weeks of age. The vertical growth of the spinal canal at C5 reached a plateau after 20 and 12 weeks, and at T8 after 20 and 16 weeks, in males and females, respectively. The vertical growth of the C5 and T8 spinal cord reached a plateau after 24 weeks in both sexes. The vertical space available for the cord (SAC) of C5 and T8 did not significantly change after 8 weeks in either sex. Western blot analyses showed that VEGFA, FGF2, and BDNF were highly expressed in the cervical spinal cords of 4-week-old rats, and that the expression of these growth factors declined as rats grew. These findings indicate that the spinal canal and the spinal cord in rats continue to grow even after sexual maturation and that rats need to be at least 8 weeks of age for use in experimental models of CSM. The present study, in conjunction with recent evidence, proposes the hypothetical model that the growth of rat spinal cord after the postnatal period is mediated at least in part by differentiation of neural progenitor cells and that their differentiation potency is maintained by VEGFA, FGF2, and BDNF.

## 1. Introduction

Cervical spondylotic myelopathy (CSM) is an age-related disease, and one of major spinal degenerative diseases in the elderly [1]. Bony spurs, a thick flavum, and bulged discs in the cervical spondylotic spine chronically compress the spinal cord, resulting in spinal cord dysfunction. Previously, a hyperostotic mouse, the tiptoe walking Yoshimura (twy) mouse, was used to unveil the pathology of CSM [2]. However, the issue that the extent of ligaments ossification varies between subjects hampers the reproducibility, i.e., creating a similarly compressed spinal cord is difficult. In addition, functional analysis was not feasible in this mouse due to multiple ankylosing joints [3]. To circumvent these issues, novel CSM models using wild-type rodents were developed. To simulate a chronically compressed spinal cord, expandable biomaterials such as synthetic aromatic polyethers and water-absorbing urethane polymers were inserted into the space between the lamina and the spinal cord, or a screw was gradually inserted through the lamina onto the spinal cord [4,5,6,7,8,9,10].

Although these CSM models have proven their usefulness when exploring the molecular pathology of CSM [11,12], one fundamental matter remains an open question: The appropriate age to start compressing spinal cord has not been determined yet. In these models, the extent of spinal cord compression depends on the space available for the cord (SAC) and the size of the space-occupying material that is inserted into the spinal canal. If the spinal canal keeps growing, the compression of the spinal cord by the inserted material will be attenuated. In humans, the size of the SAC appears to be fixed at 6–8 years of age, because the growth of spinal cord ceases at 4–5 years of age [13], and the growth in the axial size of the spinal canal stops at 6–8 years of age [14]. This is much earlier than the age at which sexual maturity is reached or when growth in height is complete [15,16]. In contrast to that, the age when the SAC in rodents is fixed remains unknown.

In spite of the fact, that whether the spinal cord is still in the developing stage or not significantly matters for its reparative capacity [17,18], the age in weeks when rat-spinal-cord growth is complete, has not been determined yet. Further, a possible molecular mechanism underlying the growth of rat spinal cord after the postnatal period remains unknown. Thus, the purposes of the present study were to obtain accurate growth curves for the spinal canal and the spinal cord in rats; to define the appropriate age in weeks for their use as a CSM model; and to propose a molecular mechanism of growth of the spinal cord after the postnatal period.

## 2. Results

### 2.1. General Body Growth

The total body length of Lewis rats continuously increased over time until the male and female rats reached 28 and 16 weeks of age, respectively (Figure 1A). Their body weight also continued to increase over time until the male and female rats reached 28 and 20 weeks of age, respectively (Figure 1B). These findings suggest that general bone growth in Lewis rats stops at 28 weeks in males and at 16 to 20 weeks in females.

### 2.2. Growth Curve of the Fifth Cervical (C5) Spine

The horizontal width of the spinal canal at C5 increased with increasing time, until the male and female rats reached 20 and 12 weeks of age, respectively. In 20-week-old male and 12-week-old female rats, it was 1.10 and 1.04 times as wide as in respective 6-week-old rats, which is when sexual function matures [19] (Figure 2A). In both males and females, the horizontal width of the spinal cord at C5 also increased with increasing time until the rats reached 20 weeks of age. In 20-week-old male and female rats, it was 1.16 and 1.17 times as wide as in respective 6-week-old rats (Figure 2B). The horizontal SAC at C5 decreased until the male and female rats reached the age of 6 and 12 weeks, respectively. After these times, no statistical difference was detected between all subsequent time points (Figure 2C), indicating that the horizontal SAC at C5 reached a plateau at 6 and 12 weeks of age in male and female subjects, respectively.

The vertical height of the spinal canal at C5 increased with time until the male and female rats reached 20 and 12 weeks of age, respectively (Figure 2D). In 20-week-old male and 12-week-old female rats, it was 1.39 and 1.20 times as large as in respective 6-week-old rats. The vertical height of the spinal cord at C5 continued to increase with increasing growth, until both males and females reached 24 weeks of age. That of males and females at 24-week-old was 1.37 and 1.33 times as large as in 6-week-old males and females, respectively (Figure 2E). The vertical SAC at C5 increased until males and females reached 8 and 6 weeks of age after which, there was no statistical difference between all subsequent time points (Figure 2F), indicating that the vertical SAC at C5 reached a plateau at 8 and 6 weeks of age in male and female Lewis rats, respectively.

### 2.3. Growth Curve of the 8th Thoracic (T8) Spine

The horizontal width of the spinal canal at T8 increased with increasing time, until both male and female rats reached 16 weeks of age. In 16-week-old male and female rats, it was 1.20 and 1.19 times as wide as in respective 6-week-old animals (Figure 3A). The horizontal width of the spinal cord at T8 also continued to increase over time, until both the male and female rats reached 20 weeks of age. In the 20-week-old rats, it was 1.37 and 1.34 times as wide as in 6-week-old males and females, respectively (Figure 3B). The horizontal SAC at T8 remained unchanged for the entire period in males, but in females it increased until they reached 6 weeks of age (Figure 3C).

The vertical height of the spinal canal at T8 increased with increasing time until the males and females reached 20 and 16 weeks of age. In 20-week-old males it was 1.38 times, and in 16-week-old females 1.27 times greater than in respective 6-week-old animals (Figure 3D). The vertical height of the spinal cord at T8 increased over time up to the age of 24 weeks in both males and females. In 24-week-old males and females, it was 1.39 and 1.31 times as large as in 6-week-old males and females, respectively (Figure 3E). The vertical SAC at T8 increased until males and females reached 6 and 8 weeks of age, respectively, and there was no significant difference between all subsequent time points that were examined (Figure 3F). This indicates that the vertical SAC at T8 reached a plateau at 6 and 8 weeks of age in males and females, respectively.

### 2.4. Histology

Safranin O staining of the C5 spinal segment revealed that an epiphyseal plate was present near the vertebral endplate at all examined time points, which were 4, 8, and 24 weeks of age (Figure 4A). Only in 4-week-old subjects, the primary ossification center was still observed in the vertebral body, but not in the lamina (Figure 4A). Hematoxylin and eosin (H&E) staining demonstrated that a thick periosteum covered the lamina in 4-week-old animals, and that the lamina contained soft tissue (Figure 4B). As the animals continued to grow, the thickness of the periosteum on the lamina decreased and the soft tissue inside the lamina disappeared. In the 24-week-old subjects, most of the soft tissue on and inside lamina was gone (Figure 4B).

### 2.5. Growth Factor Expression in Cervical Spinal Cord

To determine whether growth factors were involved with continuous growth of the adult rat spinal cord until 24 weeks of age, expression of fibroblast growth factor 2 (FGF2), vascular endothelial growth factor A (VEGFA), brain-derived neurotrophic factor (BDNF), and insulin-like growth factor1 (IGF1) in cervical spinal cords from 4-, 8-, and 24-week-old rats were analyzed using Western blot. These growth factors were reported to maintain the differentiation potency of neural progenitor cells (NPCs) in the adult rodent nervous system [20,21,22,23,24]. All of the analyzed growth factors were expressed at the highest level at 4 weeks of age among all tested ages. Band intensities of FGF2, VEGFA, and BDNF visually decreased as rats grew, whereas the band intensity of IGF1 did not significantly decline (Figure 5A,B).

## 3. Discussion

The present study revealed that the age at which the spinal canal and the spinal cord stop growing in Lewis rats depends on the sex of the animal and on the spinal level and was 20 weeks of age or older in our study, i.e., it was delayed by at least 12 weeks compared to when reproductive function matures [19]. This is significantly different from humans in whom the spinal canal and spinal cord stop growing before sexual maturation is reached [13,15]. Importantly, the vertical SAC in Lewis rats did not change significantly after 8 weeks of age, indicating that 8-week-old or older Lewis rats would be appropriate for use as a CSM model, for which a space-occupying material is inserted into the spinal canal.

General bone growth in male Lewis rats appears to be complete at 28 weeks of age, and at 20 weeks of age in females, based on the fact, that their total body length stopped increasing after these periods of time. On the other hand, the growth of the spinal canal stopped at 20 weeks of age in males and at 12 to 16 weeks of age in females, indicating that the growth of the spinal canal is completed before that of general bone growth. This is similar to humans, in which the growth of the spinal canal stops 8 to 10 years earlier than general bone-length growth is complete [25], although a precise comparison of ages between these two species is difficult. Compared to humans, rats become sexually mature at a younger age and socially mature at an older age [19,26,27].

Regarding the mechanism responsible for the growth of the spinal canal after 4 weeks of age, no epiphyseal lines were observed in bony structures of the spinal canal, which are the vertebral body, the pedicles, and the lamina. It therefore appears that the spinal canal expands via membranous ossification. In fact, periosteum tissue was present on the lamina in 4- and 8-week-old rats, whereas it was rarely detected on the lamina in 24-week-old rats, in which the spinal canal was not growing. Although the present study demonstrates that the growth curve for the body length in Lewis rats reached a plateau, the epiphyseal line next to the vertebral endplate was always present throughout the study, raising the possibility that a significant increase in body length might be detected if a longer-term observation was performed. On the other hand, no epiphyseal line remained in the spinal canal, and the periosteum was rarely present on the lamina in 40-week-old rats, indicating that it is unlikely that a longer-term observation would detect further growth of the spinal canal.

The growth curve for the spinal cord in Lewis rats reached a plateau at 24 weeks of age, which is older than the age at which sexual function matures and spinal canal growth stops [19,26,27]. This is in significant contrast to the growth curve for the human spinal cord. It stops growing at 4–5 years of age [13], which is prior to the age of sexual maturation and the cessation of spinal canal growth. This contrast could be attributed to anatomical and functional differences between rodents and humans. The spinal cord of 8- to 12-week-old Lewis rats was approximately 83–93% and 79–92% of the final size, respectively. This raises the possibility that the spinal cord in Lewis rats of these ages might possess characteristics of both developmental and mature stages. Considering that the repair and regenerative capacity of the nervous system is much greater at the developmental stage than at the mature stage [28,29], the use of growing spinal cords as disease models needs to be carefully considered, particularly when translating the data to clinical conditions. A comparative study to elucidate the spinal cord function in 8- to 12-week-old as well as 24-week-old Lewis rats is needed to increase the applicability of 8- to 12-week-old rats as a model system for human disorders.

The vertical SAC in Lewis rats did not change significantly after 8 weeks of age, indicating that, when creating a CSM model by placing a space occupying material into the sublaminar space, the use of 8-week-old or older rats would avoid the potential reduction in spinal cord compression over time. In addition, 24-week-old or older rats would be preferable for use as a CSM model, because their spinal cord does not grow in size and CSM occurs in elderly people. However, the use of such an old subject would detract from the versatility of the animal model. Therefore, further studies are needed to clarify the necessity for using old subjects to simulate the clinical aspects of CSM.

In the present study we used CTM as a method to quantify the size of the spinal canal and the spinal cord, instead of histology or magnetic resonance imaging (MRI). CTM is a clinically valuable imaging modality that can provide information that cannot be obtained with MRI [30], and there are three advantages with respect to conducting the present study. First, CTM can be used to measure the size of the spinal canal and spinal cord in a living state. For histology, specimens are usually fixed and dehydrated, making it difficult to measure the exact size in a living state. Second, in contrast to histology, there is no need to sacrifice subjects when performing CTM measurements. Lastly, CTM is superior to MRI in resolution for assessing the shape of the spinal canal and spinal cord [30]. On the other hand, CTM is disadvantageous in terms of invasiveness compared to MRI, although the injection of a contrast medium into the subdural space is simple and safe, and can be performed multiple times on the same individual. Thus, CTM is a useful imaging method for basic studies of CSM and other spinal disorders.

The present study has several limitations. The obtained results are specific for the Lewis rats at our animal facility. As general body growth in rats is influenced by congenital factors derived from each strain, as well as acquired factors such as food and living environment [31,32], other strains such as Sprague-Dawley (SD) and Wistar rats and even Lewis rats at a different facility might show different growth curves for the spinal canal and the spinal cord. Secondly, the present study did not examine the entire life of a rat, raising the possibility that the plateau of the growth curve might change if a longer period of observation was used.

Regarding the mechanism of growth of the spinal cord after the postnatal period, a differentiation from neural or glial progenitor cells could be involved in an increase in neural cells [33]. For instance, in the spinal cord of 8- to 12-week-old C57BL/6 mice, NPCs generate new neurons [34,35,36,37]. In the spinal cord of Fischer rats at 13 to 14 weeks of age, new glial cells are generated from glial progenitor cells (GPCs) [38]. Further, neurogenesis occurs in the cervical spinal cord even in 30-week-old female SD rats [39]. Accordingly, in the adult spinal cord of Lewis rats, it is reasonable to speculate that NPCs and GPCs keep generating new neural cells in sufficient number to increase the size of spinal cord until 24 weeks of age. Then later, either the number of these progenitor cells or their differentiation potency declines, resulting in a steady state of the size of spinal cord.

Concerning the molecular mechanism to maintain the differentiation potency of NPCs and GPCs in the adult rodent nervous system, several growth factors were identified by the investigation of adult hippocampi. For instance, peripheral administration of IGF1 induced neurogenesis in the adult rat hippocampus [20]. VEGF knockout mice showed reduced BrdU labeling in the subgranular zone (SGZ) of the hippocampal dentate gyrus (DG) and the forebrain subventricular zone (SVZ), and an administration of VEGFB then increased BrdU-labeled neurons in the SGZ and SVZ of adult rats [21]. Deletion of FGF receptors in NPCs in 2- to 3-month-old mice reduced the generation of new neurons in the SGZ, and activation of FGF receptor 3 increased the generation of new neurons in the SGZ [22]. Knockdown of BDNF in the DG in SD rats at 10 weeks of age decreased the number of NPCs in the DG [23]. BDNF infusion into the DG led to an increase in the generation of granule cells [24]. Of note, expression of IGF1, FGF2, and VEGF in the hippocampus in Fischer 344 rats declined during the course of aging [40]. This evidence suggests that these growth factors in the spinal cord could contribute to the continuous growth of the adult rat spinal cord.

In line with that, the current study demonstrated that expression of IGF1, VEGFA, FGF2, and BDNF in cervical spinal cords was elevated in 4-week-old rats. Later, at 8 weeks of age, expression of VEGFA, FGF2, and BDNF declined but was still greater than at 24 weeks of age. On the contrary, the expression of IGF1 in cervical spinal cords did not significantly change as rats grew. Based on these findings together with recent evidence in adult hippocampi, the current study proposes a theoretical model for the growth of the adult spinal cord in size. That is, new neural cells were generated via differentiation of NPCs, resulting in the increase in spinal cord size. The differentiation potency of these NPCs was maintained by IGF1, VEGFA, FGF2, and BDNF. As rats grew, the expression of VEGFA, FGF2, and BDNF declined, resulting in the cessation of spinal cord growth at 24 weeks of age.

This theoretical model can be verified by experiments to manipulate these growth factors. The function of these growth factors can be blocked at 8 weeks of age via local or systemic administration of functional blocking antibodies in wild-type rodents [41,42,43,44], systemic administration of tamoxifen in 8-week-old mice transgenic for tamoxifen-inducible deletion of these growth factors, or via knockout of the receptors [45,46,47]. Then, 2 or 3 months later, quantification of the size of the spinal cord could clarify the involvement of these growth factors in spinal cord growth. Similarly, the presence of these growth factors at 8 weeks of age can be ensured through their local or systemic administration in wild-type rodents, or via local injection of virus vectors such as lentivirus and adeno-associated virus carrying the genes of the growth factors into the spinal cord, which might increase the size of spinal cord. Although simultaneous manipulation of all those growth factors would conclusively prove this, tests of a single manipulations are necessary to achieve reliable results.

## 4. Materials and Methods

### 4.1. Animals

A total of 240 Lewis rats ranging from 4 weeks of age to 40 weeks of age were used. The study protocol was approved by the local ethical committee of Hokkaido University. The animals had free access to food and water throughout the study. For anesthesia, a mixture of ketamine (100 mg/kg, KETALAR^®^, Daiichi Sankyo Propharma Corporation, Tokyo, Japan) and medetomidine (0.5 mg/kg, DOMITOR^®^, Orion Corporation, Espoo, Finland) was administered via intraperitoneal injection. Total body length from muzzle to tail, and body weight were measured under anesthesia.

### 4.2. CT Myelography

Both male and female rats were subjected to CTM every 2 weeks from 4 weeks to 12 weeks of age and every 4 weeks from 12 weeks to 40 weeks of age. At each time point, ten animals per each sex underwent CTM, and one subject underwent CTM only once, to avoid potential distress and influence on growth. A two cm longitudinal incision was made over the occipital bone and the C2 process, followed by retraction of the muscles to expose the atlantooccipital ligament. After puncturing the ligament with a 27 G needle, contrast medium (Iohexol 300, Hikari Pharmaceutical Company Limited, Tokyo, Japan) was injected into the subdural space. The injected volume was determined based on body length, ranging from 20 μL to 70 μL. CT scanning was performed using a micro-CT instrument (R_mCT2, Rigaku, Tokyo, Japan) at a 20 μm isotropic resolution in a supine position around the C5 and T8 vertebrae.

### 4.3. Measurements

The CT images that were obtained were reconstructed at C5 and T8 for a coronal view perpendicular to the vertebral body. Among consecutive spinal canal CT images in the cranio–caudal direction, one image from the center of the stack was used for the measurements (Figure 6A,D). The dimensions of the spinal canal and the spinal cord were measured by means of R_mCT2 Image Analysis Software 1.2.1 (Rigaku, Tokyo, Japan), which was originally installed on the micro-CT instrument. The vertical heights of the spinal canal and spinal cord were measured at the midline, and the horizontal widths of the spinal canal and spinal cord were measured at the point of maximum length (Figure 6B,C,E,F). The average of three measurements was used for the analysis. The difference between the spinal canal and spinal cord for each subject was calculated as the SAC (Figure 6C,F).

### 4.4. Histology

4-, 8-, and 24-week-old female subjects were perfused with 4% paraformaldehyde (NACALAI TESQUE, INC., Kyoto, Japan), followed by dissection of the C5 spine. Specimens were decalcified via treatment with 23.8% formic acid for 5 days, and embedded in paraffin, which was then transversely sectioned with a section thickness of 5 µm. The sections were then stained with H&E and safranin O. Images were taken with an all-in-one fluorescence microscope (Keyence BZ-X800, Osaka, Japan).

### 4.5. Western Blotting

4-, 8-, and 24-week-old female subjects were perfused with cold saline, and cervical spinal cords at the C1-C7 spinal level were harvested. The harvested tissues were transferred to EzRIPA lysis buffer with phosphatase inhibitor at 1% concentration (ATTO corp., Tokyo, Japan) and homogenized while cooling on ice. After centrifugation (16,000 G, 20 min, 4 °C), the protein concentration in the supernatant was measured using Bradford method (TaKaRa Bradford Protein Assay Kit, Takara Bio Inc., Shiga, Japan), Then, the protein concentration of each sample was equalized to 1000 μg/mL via an addition of reverse osmosis water. Whole protein of the supernatant was separated using sodium dodecyl sulfate–polyacrylamide gel electrophoresis and transferred to a polyvinylidene fluoride membrane (Immobilon-P Membrane; Merck, Darmstadt, Germany). Following that, membranes were blocked with 5% powdered milk (<1% fat) in PBS for 1 h at room temperature, followed by overnight incubation with primary antibodies: anti-BDNF (1:2000, rabbit from GeneTex), anti-FGF2 (1:2000, rabbit from Elabscience Biotechnology), anti-VEGFA (1:100, mouse from Novus Biologicals), anti-IGF1 (1:2000, rabbit from Bioss Antibodies), and anti-GAPDH (1:5000, rabbit from GeneTex). After washing, the membrane was incubated with horseradish peroxidase-conjugated secondary antibody (1:3000, Novus Biologicals) for 1 h at room temperature. The bands were visualized using Ez WestLumi Plus (ATTO, Tokyo, Japan) and Image Lab 4.0.1 (Bio-Rad) software. Protein expression levels of the bands were quantified using ImageJ (NIH, Bethesda, Maryland, USA).

### 4.6. Statistical Analysis

The Kruskal–Wallis analysis of variance (ANOVA) with the Steel–Dwass test was used for detecting statistical differences between the different ages of the animals. If no statistical difference was detected in a measured parameter between the time points after a certain point, these ages were determined to represent a plateau in the parameter. All statistical analyses were performed with JMP software (SAS, Cary, NC, USA) with a pre-specified significance level of 95%. Data are presented as the mean ± standard deviation (SD). In Western blotting, one-way analysis of variance was performed between the three groups, and then Tukey’s post-hoc test was used. A *p*-value < 0.05 was considered statistically significant.

## 5. Conclusions

The spinal canal and the spinal cord in Lewis rats continue to grow even after sexual maturation. To create a CSM model using Lewis rats, the animals need to be least 8 weeks of age or older. The present study, in conjunction with recent evidence, proposes the hypothetical model that growth of the rat spinal cord after the postnatal period is mediated at least in part by differentiation of NPCs and that their differentiation potency is maintained by VEGFA, FGF2, and BDNF.

## Figures and Tables

**Figure 1 ijms-23-16076-f001:**
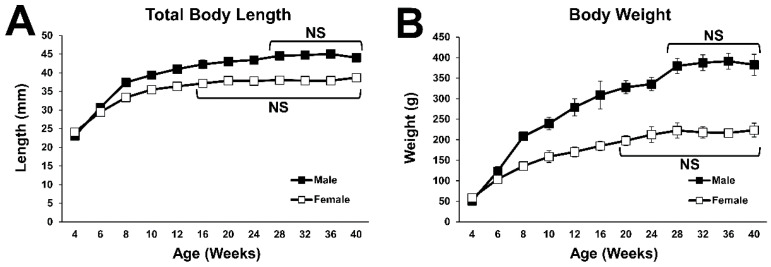
Growth curves of body size. (**A**) Growth curve for the total length from muzzle to tail. No statistical difference was detected between the time points at 28 weeks of age and older in males, and between the time points at 16 weeks of age and older in females. (**B**) Growth curve of body weight. No statistical difference was detected between the time points at 28 weeks of age and older in males, and between the time points at 20 weeks of age and older in females.

**Figure 2 ijms-23-16076-f002:**
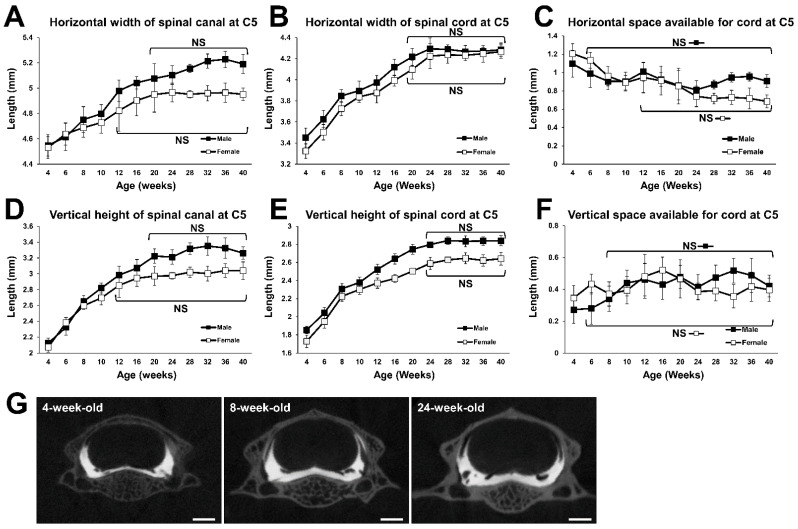
Growth curves of the spinal canal and the spinal cord at C5. (**A**) Growth curve for the horizontal width of the spinal canal. No statistical difference was detected between the time points at 20 weeks of age and older in males, and between the time points at 12 weeks of age and older in females. (**B**) Growth curve for the horizontal width of the spinal cord. No statistical difference was detected between the time points at 20 weeks of age and older in males and females. (**C**) Growth change of horizontal SAC. No statistical difference was detected between the time points at 6 weeks of age and older in males, and between the time points at 12 weeks of age and older in females. (**D**) Growth curve for the vertical height of the spinal canal. No statistical difference was detected between the time points at 20 weeks of age and older in males, and between the time points at 12 weeks of age and older in females. (**E**) Growth curve for the vertical height of the spinal cord. No statistical difference was detected between the time points for 24 weeks of age and older in males and females. (**F**) Growth change of vertical SAC. No statistical difference was detected between the time points at 8 weeks of age and older in males, and between the time points at 6 weeks of age and older in females. (**G**) Representative coronal computed tomography myelography (CTM) images of C5 spinal segment of a 4-, 8-, and 24-week-old female. Scale bars, 1 mm.

**Figure 3 ijms-23-16076-f003:**
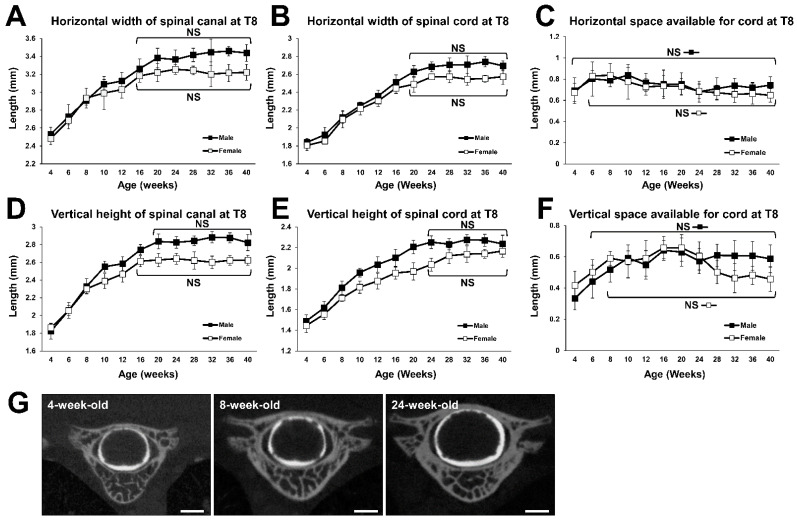
Growth curves for the spinal canal and the spinal cord at T8. (**A**) Growth curve of horizontal width of spinal canal. No statistical difference was detected between the time points at 16 weeks of age and older in males and females. (**B**) Growth curve for the horizontal width of spinal cord. No statistical difference was detected between the time points for 20 weeks of age and older in males and females. (**C**) Growth change of horizontal SAC. No statistical difference was detected in males, and between the time points at 6 weeks of age and older in females. (**D**) Growth curve for the vertical height of the spinal canal. No statistical difference was detected between the time points at 20 weeks of age and older in males, and between the time points at 16 weeks of age and older in females. (**E**) Growth curve for the vertical height of the spinal cord. No statistical difference was detected between the time points at 24 weeks of age and older in males and females. (**F**) Growth change for vertical SAC. No statistical difference was detected between the time points at 6 weeks of age and older in males, and between the time points for 8 weeks of age and older in females. (**G**) Representative coronal CTM images of T8 spinal segment of a 4-, 8-, and 24-week-old female. Scale bars, 1 mm.

**Figure 4 ijms-23-16076-f004:**
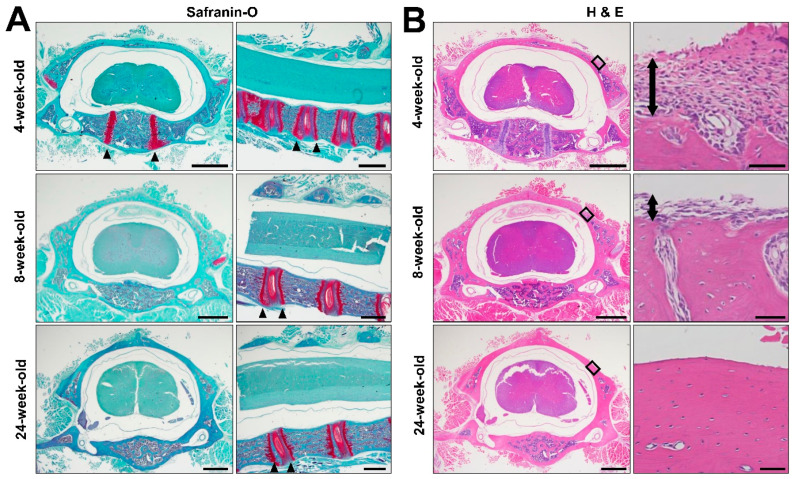
Histology of C5 Spinal segments. (**A**) Coronal and sagittal images of Safranin O-stained sections of the C5 spine in a female rat at 4, 8, and 24 week old. Arrow head indicates the primary ossification center in the vertebral body of 4-week-old subject. Epiphyseal line next to the vertebral endplate was present in all examined subjects as indicated by arrow heads. Scale bars, 1 mm. (**B**) Low and high magnification coronal images of Hematoxylin and Eosin (H&E) stained sections of C5 spinal segment in female rats at 4, 8, and 24 weeks of age. Right images are high magnification views of the boxed areas in left images. The thickness of the periosteum on the surface of the lamina decreases over time, as indicated by two-headed arrows. It finally disappeared at 24 weeks of age. Scale bars, 1 mm.

**Figure 5 ijms-23-16076-f005:**
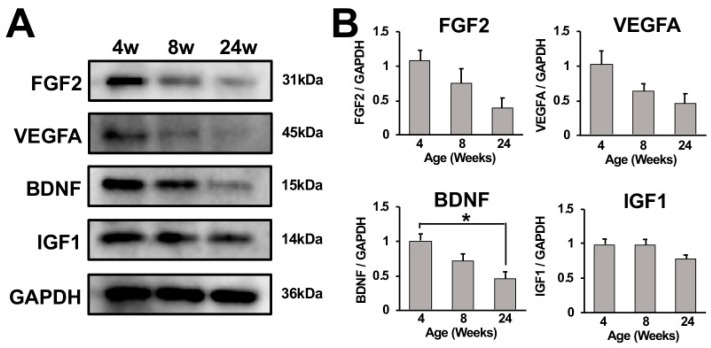
Growth factor expression in the cervical spinal cord. (**A**) Detection of FGF2, VEGFA, BDNF, and IGF1 expressions in the cervical spinal cords of a female rat at 4, 8, and 24 weeks of age using Western blot analysis. Expression of FGF2, VEGFA, and BDNF visually declined as rats grew, whereas a decrease in IGF1 expression was not apparent. (**B**) Quantification of band intensities. Results represent the mean for the relative band intensity values to GAPDH ± SEM (n = 3). The cervical spinal cord of a 4-week-old subject demonstrated significantly higher band intensity of BDNF than that of a 24-week-old subject. * *p* < 0.05. One-way ANOVA with the Tukey– Kramer test.

**Figure 6 ijms-23-16076-f006:**
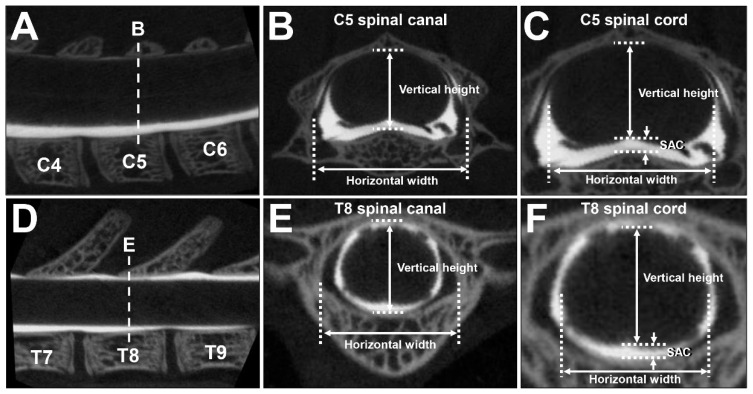
Measurement of the spinal canal and the spinal cord. (**A**–**F**) CTM images of an 8-week-old female rat. (**A**) Sagittal image of CTM of the cervical spine. Dotted line indicates the slice for which a coronal image (**B**,**C**) was measured. (**B**) Coronal image of the spinal canal at C5. Two-headed arrows indicate the vertical height and horizontal width of spinal canal for measurement. (**C**) Coronal image of the spinal cord at C5. Two-headed arrows indicate vertical height and horizontal width of spinal cord for measurement. One-headed arrows indicate vertical SAC. (**D**) Sagittal image of CTM of the thoracic spine. Dotted line indicates the slice for which a coronal image (**E**,**F**) was measured. (**E**) Coronal image of the spinal canal at T8. Two-headed arrows indicate vertical height and horizontal width of spinal canal for measurement. (**F**) Coronal image of the spinal cord at T8. Two-headed arrows indicate vertical height and horizontal width of the spinal cord for measurement. One-headed arrows indicate vertical SAC.

## Data Availability

The data generated by the current study are available from the corresponding author upon reasonable request.

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
