# Peer review of "Spinal Canal and Spinal Cord in Rat Continue to Grow Even after Sexual Maturation: Anatomical Study and Molecular Proposition"

_ijms, 2022, doi:10.3390/ijms232416076_

Round 1
Reviewer 1 Report (Previous Reviewer 3)
The authors have made significant changes to their previously rejected manuscript, and several molecular aspects were introduced, although as authors state only on descriptive level. The authors did not conduct any experiments in relation to these molecular growth factors, so in my personal opinion this manuscript is a solid anatomical study and is not without merit and has its place in the current literature, but following the opinion of fellow reviewer colleague, not in a molecular biology or medicine journal. I suggest authors to re-write and re-submit to journal of a different scope.
Reviewer 2 Report (Previous Reviewer 1)
My main concern regarding this paper was the lack of molecular data/focus for a journal like International Journal of Molecular Sciences.
I can see now that the authors have included some molecular data on section 2.5 and figure 5, and of course on the discussion. This is more reasonable and, in my opinion, improves the paper, in terms of the science itself and for this particular journal. Therefore, I am happy with this paper to be published, and I really appreciate the effort made by the authors to make the paper more suitable for the journal.
This manuscript is a resubmission of an earlier submission. The following is a list of the peer review reports and author responses from that submission.
Round 1
Reviewer 1 Report
The article by Sotome et al is a nice work presenting in a very descriptive way that both the spinal canal and cord continue to grow in a sex dependent manner even after sexual maduration. The paper is nicely presented, with all the imaging measurements well organized throughout different ages, helping out in a clear way to follow the data and reach to the conclusions.
I personally find this kind of articles very interesting and informative, as point to a perspective of sex. Traditionally most of the experimental sciences used to focus on conclusions based on experiments done in only one part of the population, so half of the population is systematically ignored, and therefore some information could be left aside.
In my opinion, the downside of this study is that indeed it is very descriptive, lacking some molecular mechanism that could help to explain or understand why the spinal cord continue to grow in adult rats. Which is something I would rather to expect in a journal of an impact of 6, as this one. The lack of some mechanism, making it very descriptive for such a journal, makes this paper very limiting.
Author Response
We really appreciate your advice and comments. Your inputs significantly improved the manuscript.
Comments:
In my opinion, the downside of this study is that indeed it is very descriptive, lacking some molecular mechanism that could help to explain or understand why the spinal cord continue to grow in adult rats. Which is something I would rather to expect in a journal of an impact of 6, as this one. The lack of some mechanism, making it very descriptive for such a journal, makes this paper very limiting.
Response:
We added two paragraphs to describe possible cellular and molecular mechanisms underlying the continuous growth of the adult spinal cord in Discussion (P7, Line 248-279), instead of actual performing molecular experiments. As mentioned in Discussion, to clarify the molecular mechanism, a series of experiments consisting of measurement of several growth factors in spinal cord, the blockade of these growth factors, and the provision of them in adult rat spinal cord is necessary. We believe that it would be another study and is out of the focus of this manuscript.
Reviewer 2 Report
Review of manuscript titled ”Spinal canal and spinal cord in rat continue to grow even after sexual maturation”. The authors have investigated the maturation of spinal canal and spinal cord in Lewis rats to identify at which age this type of rat is appropriate for cervical spondylotic myelopathy (CSM) models. Their findings are relevant to improve research in the CSM area. The figures are illustrative and references are appropriate. I have some comments.
Abstract
Please clarify in abstract that the spinal cord thickness (horizontal width?) was evaluated at C5 and T8 level.
Methods
The use of the terms “space available for the cord” (SAC) and “spinal canal” are both used, how do they differ? It seems unclear to me after reading the methods section.
Author Response
We really appreciate your advice and comments. Your inputs significantly improved the manuscript.
Comments:
Abstract, please clarify in abstract that the spinal cord thickness (horizontal width?) was evaluated at C5 and T8 level.
Response:
We modified Abstract based on the advice.
Comments:
Methods, the use of the terms “space available for the cord” (SAC) and “spinal canal” are both used, how do they differ? It seems unclear to me after reading the methods section.
Response:
According to the advice, we modified Figure 5C and F to indicate the definition of "space available for the cord (SAC)" and “spinal canal”.
Reviewer 3 Report
Authors present an animal study on 240 Lewis rats in order identify the appropriate age for cervical spondylotic myelopathy model by the elucidation of the growth curves of the spinal canal and spinal cord. CT myelography was performed on Lewis rats from 4-weeks to 40-weeks of age. The growth of spinal canal at C5 reached a plateau at 20- and 12-weeks, and that of T8 was at 20- and 16-weeks in males and females respectively.The growth of the C5 and T8 spinal cords reached a plateau at 24-weeks in both sexes; the findings indicate that spine and spinal cord continue to grow even after sexual maturation.
Whereas growth of spinal canal in humans in axial direction stops at the age of 6-8 years-way before sexual maturity, this subject was not evaluated in rodents. Since no epyphsial plates were found, it seems that membranous ossification is the mechanism of expansion. This study indicated an important issue of proper choice of rodents for the investigation of cervical spondylotic myelopathy - if the rodents were chosen which spinal cord is still developing, than its regenerative potential is far higher than following the final formation of space for the spinal cord.
However, molecular changes underpining maturation have already been published and I suggest to include this for furhter reference and comments:
McCallum-Loudeac J, Anderson G, Wilson MJ. Age and Sex-Related Changes to Gene Expression in the Mouse Spinal Cord. J Mol Neurosci. 2019 Nov;69(3):419-432. doi: 10.1007/s12031-019-01371-3. Epub 2019 Jul 2. PMID: 31267314.
I also suggest to include a literature review on current status of animal studies for investigation of spondylotic myelopathy, and if they are data on age of rodents which are included in the most important studies in the field. Also please explain what is the main ratio in these studies.
Author Response
We really appreciate your advice and comments. Your inputs significantly improved the manuscript.
Comments:
However, molecular changes underpining maturation have already been published and I suggest to include this for furhter reference and comments:
McCallum-Loudeac J, Anderson G, Wilson MJ. Age and Sex-Related Changes to Gene Expression in the Mouse Spinal Cord. J Mol Neurosci. 2019 Nov;69(3):419-432. doi: 10.1007/s12031-019-01371-3. Epub 2019 Jul 2. PMID: 31267314.
Response:
We added the suggested reference in Discussion (P7, Line 277).
Comments:
I also suggest to include a literature review on current status of animal studies for investigation of spondylotic myelopathy, and if they are data on age of rodents which are included in the most important studies in the field. Also please explain what is the main ratio in these studies.
Response:
In addition to the description of the cervical spondylotic myelopathy (CSM) animal models and identified pathologies of the CSM in Introduction, we added a paragraph to describe landmark findings about the CSM from animal studies, week ages of investigated subjects in these studies, and their appropriateness based on the current findings (P8, Line 483-494).
Round 2
Reviewer 1 Report
I really appreciate the effort made including some theoretical explanation about the possible molecular mechanism underlying the continuous growth of the adult spinal cord.
I also understand that, to clarify the molecular mechanism, further experiments would be needed, and those would be the motivation for another paper. This is why in my comment I did not ask for additional experiments to be included, but highlighted that, without molecular experiments, this paper is not a molecular study, therefore I do not see it in a molecular journal under the section molecular neurobiology.
I also understand, as the authors indicate, that rather than to a molecular study the focus of this paper is different, a mere anatomical description.
And, because all these points, my opinion remains the same.
The paper is not a molecular study (even with some theoretical molecular explanation added), therefore I personally do not see the paper suitable for a molecular journal (with the word molecular in the title), even more if it is included in the section called molecular neurobiology.
The paper is a very nice work describing anatomy mainly.
But it is not molecular, or as good as to be published in a high impact factor journal of 6.
The same way the authors think some molecular studies are not the focus of their paper, I personally think this paper, being a non-molecular study, is not under the scope of this particular molecular journal.
Author Response
First, we would like to show our appreciation to your comments and advises. We added a molecular data to our manuscript to clarify the molecular mechanism of continuous growth of adult rat spinal cord by 24-week-old. Expressions of FGF2, VEGFA, BDNF, and IGF1 in adult rat spinal cords were analyzed by Western blotting.
As mentioned in Discussion, neural progenitor cells are present in adult rodent spinal cord, and these growth factors are known to maintain the differentiation potency of neural progenitor cells in adult rodent nervous system. Therefore, sustained expression of these growth factors in adult spinal cord could contribute to the continuous growth of the adult rat spinal cord.
Our result showed that expressions of all growth factors at 4-week-old were greater than those at 8- and 24-week-old and that expressions of FGF2, VEGFA, and BDNF apparently decreased as rats grow, suggesting that these growth factors could be involved with the continuous growth of adult spinal cord by 24-week-old.
To make a concrete conclusion, the blockade of their functions and their provision in adult rat spinal cord needs to be done. However, we believe that this information significantly increases the scientific value of the current study, since this is the first study to analyze age-related expressions of these growth factors in adult rodent spinal cord and to relate their expressions with adult spinal cord growth.
Reviewer 3 Report
According to remarks of other fellow reviewers, this manuscript does not seem to be a molecular study, i.e. it seems that no molecular experiments were done, but just a theoretical and anatomical explanation. I suggest either to perform these experiments or to change the entire title of the manuscript into a theoretical-anatomical model/proposition, otherwise I do not suggest it for publishing.
Author Response
First, we would like to show our appreciation to your comments and advice. As suggested, we performed a molecular experiment. And this response is same as the one to Reviewer 1.
We added molecular data to our manuscript to clarify the mechanism of continuous growth of adult rat spinal cord by 24-week-old. Expressions of FGF2, VEGFA, BDNF, and IGF1 in adult rat spinal cords were analyzed by Western blotting.
As mentioned in Discussion, neural progenitor cells are present in adult rodent spinal cord, and these growth factors are known to maintain the differentiation potency of neural progenitor cells in adult rodent nervous system. Therefore, sustained expression of these growth factors in adult spinal cord could contribute to the continuous growth of adult rat spinal cord.
Our result showed that expressions of all growth factors at 4-week-old were greater than those at 8-and 24-week-old and that expressions of FGF2, VEGFA, and BDNF apparently decreased as rats grow, suggesting that these growth factors could be involved with the continuous growth of adult spinal cord by 24-week-old.
To make a concrete conclusion, the blockade of their functions and their provision in adult rat spinal cord needs to be done. However, we believe that this information significantly increases the scientific value of the current study, since this is the first study to analyze age-related expressions of these growth factors in adult rodent spinal cord and to relate their expressions with adult spinal cord growth.